# The Influence of COPD Awareness on Hospital Admissions: A Paradoxical Relationship? [note 1]

**DOI:** 10.3390/healthcare13121438

**Published:** 2025-06-16

**Authors:** Deniz Çelik, Murat Yıldız, Oral Menteş, Özkan Yetkin, Hüseyin Lakadamyalı, Savaş Gegin, Ahmet Yurttaş, Maşide Arı, Derya Kızılgöz, Kerem Ensarioğlu, Afife Büke

**Affiliations:** 1Department of Pulmonology, Faculty of Medicine, Alaaddin Keykubat University, 07490 Antalya, Turkey; drdenizcelik@hotmail.com (D.Ç.); ozkanyetkin@hotmail.com (Ö.Y.); lakadamyali.h@gmail.com (H.L.); uutfahmetyurttas@gmail.com (A.Y.); 2Department of Pulmonology, Atatürk Sanatorium Research Hospital, Faculty of Medicine, Health Sciences University, 06290 Ankara, Turkey; omentes@live.com (O.M.); masidetuten@icloud.com (M.A.); deryaozaydin01@hotmail.com (D.K.); kerem.ensarioglu@gmail.com (K.E.); afifebuke@icloud.com (A.B.); 3Department of Pulmonology, Samsun Training and Research Hospital, 55090 Samsun, Turkey; geginn@hotmail.com

**Keywords:** COPD knowledge, hospital readmissions, Bristol COPD Knowledge Questionnaire, healthcare utilisation

## Abstract

**Background:** Chronic obstructive pulmonary disease (COPD) is a progressive respiratory condition characterised by frequent exacerbations, which contribute to increased healthcare utilisation and reduced quality of life. Knowledge about the disease is generally associated with better outcomes. This study examined the association between COPD knowledge levels and healthcare utilisation (including hospital readmissions) in patients hospitalised for acute exacerbations. **Methods**: This prospective observational study included 78 patients hospitalised for COPD exacerbations and classified as Group D according to the updated GOLD criteria 2021. The Bristol COPD Knowledge Questionnaire (BCKQ) was administered prior to discharge to evaluate patients’ knowledge levels. Data were collected about emergency department visits, hospitalisations, and intensive care unit (ICU) admissions for a six-month follow-up period. Statistical analyses assessed the relationships between BCKQ scores, patient outcomes, and risk factors influencing hospital readmissions. **Results**: The median BCKQ total score was 23 (6–40). A strong correlation was found between higher BCKQ scores and more visits to the emergency room (*p* = 0.005), especially in the subdomains of epidemiology (*p* = 0.010), aetiology (*p* = 0.033), and dyspnoea (*p* = 0.042). Higher antibiotic knowledge scores were associated with ICU admissions (*p* = 0.019). Logistic regression analysis revealed that domiciliary NIV use (OR = 2.60, *p* = 0.041) and higher BCKQ scores (OR = 1.10, *p* = 0.010) were significant predictors of hospital readmissions. However, no significant relationship was found between survival and BCKQ or mCCI scores (*p* > 0.05). **Conclusions**: This study indicates that while increased COPD knowledge is associated with greater healthcare utilisation, it does not directly translate into improved clinical outcomes. These findings underscore the importance of integrating practical skills and behaviour management into educational programmes to help patients effectively apply their knowledge. Further research is needed to explore long-term implications and strategies to optimise knowledge-based interventions.

## 1. Introduction

Chronic obstructive pulmonary disease (COPD) is among the leading causes of chronic morbidity and mortality worldwide. The disease is characterised not only by persistent airflow limitation but also by a spectrum of systemic effects and exacerbation patterns that require frequent interactions with the healthcare system. As COPD progresses, patients may experience an increasing number of acute episodes that disrupt their daily lives and often necessitate urgent medical attention.

COPD is a progressive respiratory condition characterised by frequent exacerbations, emergency department visits, and hospitalisations, all of which contribute to a substantial global health burden. These acute exacerbations not only diminish patients’ quality of life but also significantly increase the risk of all-cause mortality, particularly in individuals with moderate to severe COPD. Evidence suggests that frequent exacerbations accelerate lung function decline, intensify symptom severity, and lead to greater reliance on healthcare resources [1,2]. High readmission rates following hospitalisations further exacerbate this burden, creating an unsustainable strain on healthcare systems. For patients, recurrent hospitalisations are both physically taxing and emotionally distressing, often reflecting suboptimal disease management. As such, identifying the factors that contribute to exacerbations is crucial for improving patient outcomes and reducing healthcare costs [3]. While pharmacologic treatments and guideline-directed care are central to managing COPD, recent studies underscore the critical role of patient knowledge and self-management capacity in determining outcomes. Patients who better understand their disease are more likely to adhere to maintenance therapy, recognise early warning signs of exacerbation, and seek timely care. However, existing literature also indicates substantial variability in COPD patients’ baseline knowledge, and how this translates into clinical outcomes remains poorly understood [4,5,6]. Evaluating COPD patients’ current level of knowledge about their disease is a vital step toward developing effective educational programmes. This study aims to determine whether the baseline knowledge levels of patients—specifically those who have not received prior intervention or education regarding COPD—are associated with exacerbation frequency, hospital readmissions, or hospitalisation rates. Previous research highlights variability in patients’ understanding of their condition, which may influence health outcomes [7,8,9,10,11]. By exploring the relationship between knowledge levels and exacerbation frequency, this study seeks to identify critical knowledge gaps that may contribute to suboptimal disease management. Gaining insights into these baseline knowledge deficits will be instrumental in designing tailored educational interventions aimed at reducing exacerbations and improving the overall management of COPD.

Unlike many previous studies that focused on patients already enrolled in structured education programmes or pulmonary rehabilitation, our study uniquely evaluates patients at a point when no formal education has yet been delivered. This design offers a clearer lens to investigate how unassisted knowledge alone may relate to healthcare utilisation in real-world settings. Our findings may inform the development of more targeted, personalised interventions that align with actual patient needs and learning baselines [12].

## 2. Materials and Methods

This study was conducted in accordance with the Declaration of Helsinki and was approved by the Keçiören Training and Research Hospital Clinical Research Ethics Committee (approval date: 13 April 2021; decision number: 2012-KEAK 15/2245). Informed consent was obtained from all patients, both in written and verbal form, with nonverbal communication being considered an exclusion criterion.

### 2.1. Study Design

This prospective, observational study was designed to evaluate the impact of knowledge levels on readmission and hospitalisation rates among patients with COPD. As a non-interventional study, it utilised questionnaires administered to a defined patient cohort, followed by a minimum follow-up period of six months. The study adheres to the Strengthening the Reporting of Observational Studies in Epidemiology (STROBE) guideline.

### 2.2. Study Group and Inclusion Criteria

The study population comprised patients diagnosed with COPD who were treated in the Chest Diseases Intensive Care Unit and the Chest Diseases Clinics of our tertiary thoracic diseases training and research hospital. A total of 146 patients were screened. Due to exclusion criteria, 68 patients were excluded. The sample size consisted of 78 patients. Clinical characteristics were retrieved from patient records and the hospital’s data processing system, with all relevant information recorded in a standardised data collection form. Prior to discharge, participants completed the Bristol COPD Knowledge Questionnaire (BCKQ). After BCKQ patients were discharged, the follow-up period started. For both ward and ICU admissions, only hospitalisations due to COPD exacerbations were evaluated. Additionally, only emergency department visits related to COPD-related symptoms were considered and included in the study.

The study employed relatively strict exclusion criteria, such as the omission of patients with significant comorbid respiratory or neurological conditions, to minimise confounding and ensure a more homogeneous sample focused specifically on COPD-related hospitalisations. While this approach strengthens internal validity and enhances the clarity of observed associations, it may limit the external generalisability of the findings to the broader COPD population, where multimorbidity is common. Nonetheless, this methodological choice was necessary to isolate the effects of disease-specific knowledge and avoid misinterpretation due to overlapping pathologies. Future studies may benefit from a more inclusive design that reflects real-world clinical diversity while still controlling for key confounders through statistical adjustments.

### 2.3. Inclusion Criteria

As the study population was composed entirely of GOLD Group D patients according to the classification available at the time of data collection (2021), no reclassification was performed under the newly updated GOLD 2023 system. In the new classification, Group D is included in Group E. However, we acknowledge the shift in GOLD’s emphasis toward symptom burden and exacerbation risk rather than ABCD grouping. This paradigm shift further reinforces the need for integrated approaches targeting both clinical and behavioural dimensions of COPD management. Future studies might benefit from stratifying patients using newer multidimensional indices that align with the updated GOLD criteria to better capture phenotypic variability and tailor interventions accordingly.

Patients diagnosed with COPD and classified as Group D according to the updated Global Initiative for Chronic Obstructive Lung Disease (GOLD) criteria.Patients hospitalised due to COPD exacerbation.Patients who signed the informed consent form.

### 2.4. Exclusion Criteria

Patients with neurological diseases (e.g., Alzheimer’s disease, Parkinson’s disease) or other forms of dementia were excluded.Patients with congestive heart failure.Patients with asthma, obesity hypoventilation syndrome, or obstructive sleep apnoea syndrome.Patients with cerebrovascular diseases or sequelae of such conditions.Patients with restrictive lung diseases.Patients with musculoskeletal or joint disorders causing immobility.Patients residing beyond the central districts of Ankara were excluded.Patients with incomplete data or missing file information.Patients who declined to participate in the study.Patients with a positive coronavirus disease 2019 (COVID-19) PCR test or those receiving COVID-19 specific treatment.Patients diagnosed with any malignancy.

### 2.5. Bristol COPD Knowledge Level Questionnaire (BCKQ)

The Bristol COPD Knowledge Level Questionnaire (BCKQ) is a tool developed to evaluate the knowledge level of patients with COPD regarding their condition. It assesses patients’ general understanding of COPD, as well as their knowledge on various topics such as disease symptoms, treatment approaches, respiratory techniques, and overall disease management.

The BCKQ comprises a total of 65 questions, each requiring a response of “yes”, “no”, or “not sure”. Correct answers are awarded one point, while incorrect or uncertain answers are scored as zero. The cumulative score reflects the patient’s level of knowledge about COPD. The questionnaire is organised into 13 subcategories, each addressing a specific domain of COPD knowledge. These subcategories include ‘epidemiology’, ‘symptoms’, ‘shortness of breath’, ‘sputum’, ‘exercise’, ‘smoking’, ‘vaccines’, ‘inhaled bronchodilators’, ‘antibiotics’, and ‘oral steroids’. Each section is designed to evaluate distinct aspects of disease understanding and to identify areas where patients may lack critical information for effective disease management. The total score ranges from 0 to 65, with higher scores indicating greater knowledge and lower scores suggesting insufficient awareness of COPD [13].

In this study, the BCKQ was administered to patients prior to discharge using a face-to-face interview method. Permission to use the original questionnaire in its original language was obtained from Mr. Roger White, who developed the BCKQ and conducted its validation study [9]. Additionally, approval to use the Turkish version of the BCKQ was secured from Mr. Sadık Hançerlioğlu, who successfully validated the questionnaire for use in Turkish [14].

The administration of the Bristol COPD Knowledge Questionnaire (BCKQ) was conducted in a face-to-face format by trained healthcare professionals prior to patient discharge. On average, patients completed the questionnaire in approximately 15 to 20 min, depending on their cognitive capacity and level of engagement. All sessions were conducted in a quiet setting to minimise distractions and ensure accurate responses.

### 2.6. Data Collection

We obtained demographic data, such as age, gender, smoking status, and COPD stage, by interviewing the patient or obtained the data from patients’ files and the hospital’s data recording system. The frequency of exacerbations experienced by patients over the past year, as well as emergency department visits and hospitalisations related to these exacerbations, was collected retrospectively. Following discharge, patients were monitored for six months via telephone follow-up, face-to-face interviews, or through the hospital’s data recording system, with readmissions and hospitalisations being recorded during this period.

Patients hospitalised for COPD exacerbation were identified through hospital admission logs and approached during their inpatient stay. After obtaining informed consent, baseline demographic data and clinical characteristics were collected through structured interviews and medical record review. The BCKQ was administered in person by the research team prior to discharge. Follow-up data on emergency visits, readmissions, and ICU stays were obtained from the hospital information system and supplemented by scheduled telephone interviews conducted at three and six months post-discharge.

### 2.7. Anonymity and Confidentiality

Ethical approval for this study was obtained from the institutional review board of Keçiören Training and Research Hospital Clinical Research Ethics Committee (approval date: 13 April 2021; decision number: 2012-KEAK 15/2245), and all participants provided written informed consent. Patient anonymity and confidentiality were strictly maintained throughout the study. Data were coded and stored in a secure, password-protected electronic database accessible only to authorised study personnel. Identifiable information was removed prior to data analysis to ensure compliance with data protection regulations and ethical standards.

### 2.8. Statistical Analysis

We used SPSS version 22.0 statistical software to analyse the obtained data. Categorical variables are expressed as the frequency and percentage, ordinal or non-normally distributed numerical variables as the median (minimum–maximum), and normally distributed numerical variables as the mean ± standard deviation (mean ± SD). Relationships between COPD knowledge level and readmission or hospitalisation were evaluated using the chi-square or Fisher’s exact test for categorical variables. For numerical variables, Student’s *t*-test or the Mann–Whitney U test is performed. A *p*-value of <0.05 was considered statistically significant. ROC analysis was performed using MedCalc Statistical Software version 23.2.1 (MedCalc Software Ltd., Ostend, Belgium), providing AUC, 95% confidence intervals, *p*-values, and optimal cut-off values.

A post-hoc power analysis was performed using the software G*Power version 3.1.9.7 to evaluate the statistical sensitivity of the logistic regression model employed in this study. Based on an observed effect size (odds ratio), a significance level (α) of 0.05, and a total sample size of 78, the calculated statistical power (1 − β) exceeded 0.80, indicating an acceptable probability of detecting true effects. While no a priori sample size estimation was conducted due to the observational design, the post-hoc analysis supports the adequacy of the sample size for the primary outcome variables examined.

## 3. Results

A total of 78 patients were included in the study, of whom 79.5% (62 patients) were male and 20.5% (16 patients) were female. The demographic characteristics of these patients are summarised in Table 1.

The majority of the patients had completed primary school (48.7%). Most of the patients were independent and living in their own houses. We can assume that patients may make the majority of the decisions about readmission to the hospital.

Table 2 displays the distribution of the patients’ comorbidities. Despite numerous comorbidities being designated as exclusion criteria, it was unexpected that comorbid disorders not included among the exclusions were prevalent in a significant proportion of the patients (92.3%). The most common comorbidities were hypoxic respiratory failure, hypertension, and coronary artery disease.

After a follow-up period of six months, the total scores and subscale scores of the Bristol COPD Knowledge Questionnaire (BCKQ), as well as the modified Charlson Comorbidity Index (mCCI) scores, were analysed. The aim was to evaluate potential relationships between the number of emergency department visits, hospitalisations (including intensive care unit admissions), and the survival status of the patients at the end of the follow-up period. The results of this analysis are presented in Table 3.

There was a statistically significant correlation between the total scores on the BCKQ and the sub-dimensions of epidemiology (*p* = 0.010), aetiology (*p* = 0.033), dyspnoea (*p* = 0.042), and inhaled steroids (*p* = 0.048) in patients with emergency room readmission. It was noted that patients with higher BCKQ total scale scores were more likely to be admitted to the emergency department, and this relationship was statistically significant. This finding represents a key contribution of the study. Patients with and without ward hospitalisations showed a statistically significant relationship in the symptoms sub-dimension of the BCKQ scale (*p* = 0.042). A higher symptoms knowledge score was associated with hospitalisation in the wards.

The relationship between intensive care unit (ICU) hospitalisation and the mCCI and/or BCKQ scale scores in the first six months after discharge was also evaluated. A statistically significant correlation was found between the antibiotics sub-dimension of the BCKQ scale and the number of ICU admissions (*p* = 0.019). Patients with higher antibiotic knowledge scores were more likely to be hospitalised in the ICU. At the end of the follow-up, the relationship between survival and the mCCI and/or BCKQ scale scores was assessed. Table 3 revealed no statistically significant correlation between patient survival and the mCCI or BCKQ scale scores (*p* > 0.05 for all comparisons).

The results of the logistic regression analysis, which examined the risk factors affecting hospital admissions in the first six months after discharge, are presented in Table 4. It was observed that patients’ use of domiciliary NIV devices, BCKQ total scores, and other health problems like urogenital diseases had a significant effect on their decision regarding hospital admission. Specifically, patients who used domiciliary NIV devices were 2.6 times more likely to be admitted to the hospital than those who did not use such devices (*p* = 0.041). Additionally, each unit increase in patients’ BCKQ total scores was associated with a 1.1-fold increase in hospital admission likelihood (*p* = 0.010) (Table 4).

In the ROC analysis conducted to evaluate the predictive performance of the total Bristol COPD Knowledge Questionnaire (BCKQ) score in relation to emergency department visits within six months after discharge, a cut-off value of 21 was identified. At this threshold, the sensitivity was calculated as 77.5%, specificity as 57.89%, positive predictive value (PPV) as 65.95%, and negative predictive value (NPV) as 70.97% (*p* = 0.0036, AUC = 0.683, 95% CI: 0.568–0.784) (Figure 1).

Because the ROC analyses for the BCKQ total score did not yield statistically significant curves for the ward admissions or ICU admissions, these outcomes were not included in the final analysis. As shown in Table 3, the BCKQ total score was only significantly associated with emergency department visits. No specific cut-off value for the BCKQ total score has been identified in the existing literature.

## 4. Discussion

The main finding of this study is that, contrary to expectations, increased knowledge in COPD patients was associated with higher rates of emergency department visits. Considering that increased knowledge is generally associated with better health outcomes in the literature, this result is noteworthy and suggests the need for a deeper understanding of the factors influencing patient behaviour.

COPD, a significant public health issue worldwide, remains poorly understood by the general population. Consequently, healthcare institution admissions are insufficient, and many COPD cases remain undiagnosed. Even among diagnosed individuals, symptoms are often poorly managed due to a lack of adequate disease knowledge [15].

In a study conducted by Ma et al. with 175 internal medicine nurses, including respiratory (n = 75) and non-respiratory nurses (n = 100), the overall BCKQ score was 35.76 ± 5.49 out of a maximum score of 65 [16]. Interestingly, when this score was compared with the scores of patients with no prior training, as reported by White et al. (36.1 ± 5.9) and Wong et al. (40.28 ± 10.76), the nurses’ knowledge appeared to be lower than that of these patients, indicating a deficiency in COPD knowledge among healthcare professionals [11,13]. In contrast, in a South Korean study involving 245 patients, Lee et al. reported a BCKQ total score of 28.1 ± 7.4 [17]. In our study, the median BCKQ score was 23, which is lower than those reported in other studies. This indicates significant differences in patient knowledge levels across countries and highlights a notably lower knowledge level among COPD patients in Turkey.

In Turkey, the absence of a strict referral chain within the healthcare system allows patients to directly access secondary or tertiary care, particularly specialists such as pulmonologists. Many patients prefer to bypass primary care, believing that a specialist is more likely to understand their condition accurately. Additionally, diagnostic and treatment capacities at the primary care level remain limited in many regions, leading to a reliance on hospital-based care. Regarding self-management, patients were prescribed home-based treatments such as nebulisers, long-term oxygen therapy (LTOT), or non-invasive ventilation (NIV) when clinically indicated. Moreover, they were encouraged to acquire pulse oximeters for home monitoring. Prior to discharge, all patients received brief face-to-face training on managing acute exacerbations at home, which included recognising warning signs and adjusting treatments accordingly.

The literature frequently emphasises the positive effects of increased knowledge on disease management. For instance, Collinsworth et al. demonstrated that training about COPD shortened the time to first rehospitalisation for all causes, including COPD [18]. A study conducted in China found that low levels of COPD knowledge (as measured by the BCKQ) were associated with anxiety, depression, reduced functional capacity, and poorer quality of life [19]. However, Milner et al. observed that efforts to increase participation in pulmonary rehabilitation programmes were not directly linked to COPD knowledge levels; despite being informed, patient participation rates remained low [20]. Similarly, Henoch et al., in a Swedish longitudinal registry study, found that non-participation in educational programmes did not influence exacerbation or hospitalisation rates but was associated with a significant decline in health-related quality of life (HRQoL) [21].

Unlike some of these findings, our study suggests that increased knowledge of COPD does not always lead to better outcomes. Among our patient population, higher knowledge levels appeared to make patients more aware of their symptoms, potentially driving them to seek healthcare more frequently. This indicates that awareness alone does not always result in better decision-making.

In our study, patients with higher scores in the antibiotic subdomain of the BCKQ were less likely to be admitted to the intensive care unit (ICU) within six months following discharge. This finding suggests that improved knowledge regarding antibiotic use may play a protective role by enabling patients to recognise infection-related symptoms earlier and seek timely outpatient care, thereby preventing progression to severe exacerbations. This aligns with the 2021 Global Initiative for Chronic Obstructive Lung Disease (GOLD) guidelines, which recommend prescribing antibiotics for patients exhibiting increased sputum purulence alongside dyspnoea or changes in sputum volume, emphasising the importance of early recognition and treatment of bacterial infections to prevent severe outcomes. Furthermore, a retrospective study by Vanoverschelde et al. (2023) found that in-hospital antibiotic use among patients with severe acute exacerbations of COPD was associated with longer hospital stays, highlighting the significance of appropriate antibiotic management in influencing patient outcomes. These findings support the hypothesis that focused knowledge, particularly on infection management, can have a tangible impact on disease trajectory and healthcare utilisation in COPD [22].

In line with our findings, Lewis et al. showed that while educational programmes improved self-management skills, they were insufficient for alleviating the disease burden [9]. Additionally, Zhang et al. found that while low knowledge levels were linked to higher rates of anxiety and depression, increased knowledge could also exacerbate psychological distress in certain patients [2]. In our study, increased symptom awareness may have contributed to increased emergency department visits, potentially driven by heightened anxiety. This underscores the need to consider the psychological effects of increased knowledge on healthcare utilisation.

Our results showed that patients with higher BCKQ scores had significantly more emergency department visits than those with lower scores. This finding suggests that increased knowledge is not a stand-alone factor influencing clinical outcomes. Psychological factors such as anxiety, depression, and disease burden likely play a significant role in shaping patient behaviour. Additionally, how patients interpret and apply their knowledge in practice is critical. Recognising symptoms earlier may lead to more frequent healthcare utilisation, highlighting the need for strategies to integrate this behaviour into the healthcare system effectively.

Although increased disease knowledge is generally considered beneficial, our findings support the growing consensus that knowledge alone may be insufficient for producing meaningful behaviour change in chronic disease management. This discrepancy highlights the so-called “knowledge–behaviour gap” in COPD care. Patients may be aware of appropriate actions but still fail to implement them due to psychological, cognitive, or social barriers. As such, future interventions should not only focus on education but also incorporate behavioural change techniques, such as motivational interviewing, goal setting, problem-solving training, and practical self-management skill workshops. Additionally, emotion regulation strategies, such as stress management or coping skills training, may be especially useful for patients who experience anxiety in response to symptom exacerbation. Embedding such elements into structured pulmonary rehabilitation or nurse-led follow-up programmes could enhance patients’ ability to translate knowledge into action, ultimately improving long-term outcomes.

Wong and Yu’s study demonstrated that demographic factors influence patients’ knowledge levels, with lower educational attainment associated with higher COPD knowledge levels [11]. This points to inadequate health literacy and underscores the need for tailored patient education programmes. Specifically, older patients and those with lower education levels may require additional support to effectively utilise their knowledge. Education programmes should not only focus on imparting knowledge but also on helping patients translate this knowledge into actionable self-management strategies.

Ferrone et al. found that integrated disease management (IDM) programs improved knowledge and clinical outcomes in high-risk COPD patients. However, even in this study, increased knowledge did not significantly influence direct clinical outcomes [1]. Similarly, Ward et al. emphasised that while educational programmes enhanced patient satisfaction, they did not achieve the desired success in symptom control [3]. These findings align with our study results, suggesting that increased knowledge alone may not guarantee better clinical outcomes.

Choi et al. examined the impact of adherence to action plans and knowledge levels on health outcomes in COPD patients and found that higher knowledge levels were associated with better outcomes. However, the study highlighted that the correct application of knowledge was critical [23]. Hill et al. similarly observed that short-term education programs improved knowledge levels but had limited effects on disease management without ongoing support [24]. Despite higher knowledge levels, our study found increased emergency visits, suggesting patients struggle to apply knowledge effectively.

While Choi et al. reported a positive correlation between disease-specific knowledge and improved clinical outcomes in COPD patients, our findings revealed a seemingly paradoxical trend, wherein higher knowledge scores were associated with increased healthcare utilisation. This discrepancy may be attributed to contextual and systemic differences. For instance, Choi’s study was conducted in a structured healthcare environment with well-integrated COPD action plans and follow-up systems, which likely facilitated the translation of knowledge into appropriate behaviour. In contrast, in settings where access to continuity of care, self-management training, or primary care support is more fragmented—as is often the case in resource-limited health systems—patients may resort to emergency services more frequently despite having sufficient knowledge. Additionally, cultural differences in health-seeking behaviour, trust in outpatient services, and perceived urgency of symptoms may influence how knowledge is applied. These contextual factors should be considered when interpreting cross-cultural comparisons in COPD management research [23].

Studies examining the relationship between self-management skills and knowledge levels in COPD patients point to a complex interaction. Lee et al. reported that increased knowledge could improve self-sufficiency, which in turn positively impacts health outcomes. However, they also noted that increased knowledge alone was insufficient in patients with low self-sufficiency, who required additional supportive interventions [10]. This may explain the increased emergency department visits observed in our study; while patients may have knowledge, they may lack the skills or confidence to utilise it effectively.

Our study has some limitations. One of the acknowledged limitations of this study is the relatively small sample size (n = 78). While this number may seem modest, a post hoc power analysis confirmed that the sample was statistically adequate to detect meaningful differences in the studied outcomes. Nevertheless, given the clinical and behavioural heterogeneity of COPD patients, a larger cohort could have enhanced the robustness and external validity of the findings. Future studies with broader inclusion may provide more generalisable insights and enable subgroup analyses that were not feasible within the current sample size. Additionally, the follow-up period of six months was relatively short; however, for patients with moderate to severe COPD (Group D), this timeframe is reasonable given the frequency of exacerbations. In fact, because this group of patients have frequent exacerbations, the relationship between knowledge level and readmission will be lost because of excessive exacerbations that will occur in longer or very long follow-up periods.

In our regression analysis, we utilised the enter method to include variables based on both clinical relevance and theoretical considerations. As all participants were classified as GOLD Group D, the severity of COPD was consistent across the cohort, thereby minimising variability in disease stage as a potential confounder. Variables included in the model—such as age, comorbid conditions (as measured by the mCCI), use of domiciliary non-invasive ventilation (NIV), and long-term oxygen therapy—were selected due to their established association with hospital readmission risk. Additionally, knowledge-related parameters, including the total BCKQ score, were incorporated to address the primary objective of the study. Although no formal multivariable screening process (e.g., stepwise selection) was used, care was taken to construct a model that accounted for key clinical and behavioural contributors without overfitting.

Given the exploratory nature of this study, the associations observed between BCKQ subdomains and clinical outcomes (including emergency visits, general ward admissions, and ICU admissions) should be interpreted with caution. Although several subdomains reached nominal significance (*p* < 0.05), none remained statistically significant after applying a conservative Bonferroni correction. Therefore, these findings should be considered hypothesis-generating and warrant further investigation in larger, confirmatory studies.

Future studies should aim to assess not only disease-specific knowledge but also patient adherence to COPD therapies, as knowledge alone may not ensure effective disease self-management. Integrating validated adherence measures could help uncover whether higher knowledge levels translate into better treatment compliance and reduced healthcare utilisation. Such multidimensional approaches may offer deeper insights into the behavioural pathways influencing clinical outcomes in COPD.

## 5. Conclusions

In conclusion, this study highlights that increasing knowledge alone is insufficient and does not have a direct impact on clinical outcomes in COPD patients. While prior studies, such as the one conducted by Choi et al., have demonstrated a positive relationship between treatment adherence and knowledge levels—indicating that patients with higher knowledge achieve better health outcomes [23]—our findings reveal a more complex relationship. Specifically, increased knowledge does not always result in the anticipated clinical improvements and, in some cases, may even lead to unnecessary healthcare utilisation.

These results underscore the need for health professionals to not only focus on increasing patients’ knowledge but also emphasise teaching them how to effectively apply this knowledge to manage their disease. Comprehensive and supportive education programmes that integrate practical skills training and behaviour management strategies are essential to achieving better outcomes.

Future research should explore the long-term consequences of increased knowledge, particularly its impact on patient behaviour and healthcare utilisation patterns. Understanding how to translate knowledge into effective disease management remains a critical area for further investigation.

## Figures and Tables

**Figure 1 healthcare-13-01438-f001:**
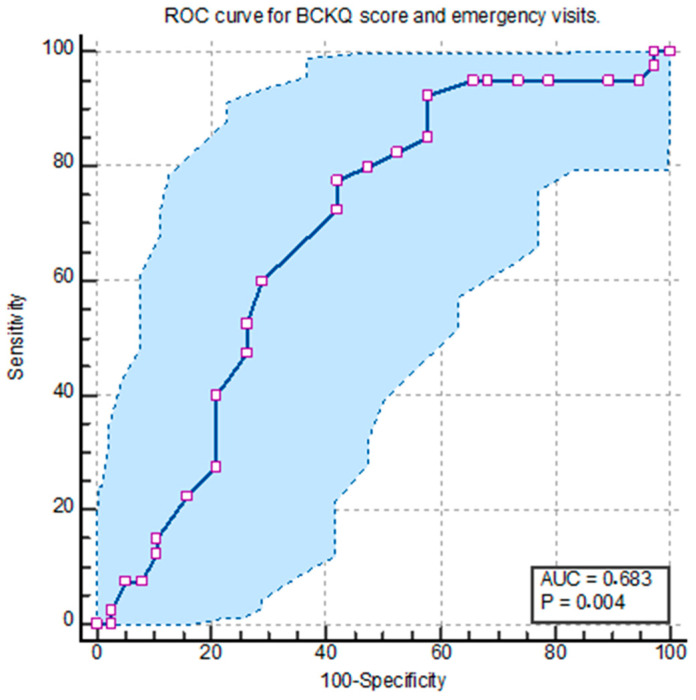
Receiver operating characteristic (ROC) curve for BCKQ score and emergency visits.

**Table 1 healthcare-13-01438-t001:** Distribution of demographic characteristics of study population.

Characteristics	n (%) or Median (Min–Max)
Age, year	70 (40–89)
Gender	
Male	62 (79.5)
Female	16 (20.5)
COPD disease age, years	9.5 (0.1–35)
Home oxygen device	56 (71.8)
Home NIV device	36 (46.2)
Education status	
Illiterate	4 (5.1)
Literate	7 (9)
Primary School	38 (48.7)
Middle School	12 (15.4)
High School	11 (14.1)
University	5 (6.4)
Master’s degree	1 (1.3)
Cared patient	19 (24.4)
Partially dependent	19 (100)
Fully dependent	0 (0)
Caregiver	
First-degree relative	18 (94.7)
Paid carer	0 (0)
Other	1 (5.3)
Place of residence	
Own house	57 (73.1)
House of relatives	20 (25.6)
Nursing home	1 (1.3)
Latest status	
In life	70 (89.7)
Exitus	8 (10.3)

**Table 2 healthcare-13-01438-t002:** Distribution of comorbidities among the participants.

Characteristics	n (%) or Median (Min–Max)
Any comorbidity	72 (92.3)
Hypoxic respiratory failure	43 (55.1)
Hypertension	38 (48.7)
Coronary artery disease	22 (28.2)
Diabetes mellitus	20 (25.6)
Benign prostatic hyperplasia	18 (23.1)
Hypercapnic respiratory failure	9 (11.5)
History of pulmonary thromboembolism	11 (14.1)
Atrial fibrillation	4 (5.1)
Chronic kidney disease	6 (7.7)
Hyperlipidemia	5 (6.4)
Bronchiectasis	5 (6.4)
History of pulmonary tuberculosis	5 (6.4)
Thyroid diseases	2 (2.6)
Non-malignant haematological disease	1 (1.3)
Rheumatoid arthritis	1 (1.3)

**Table 3 healthcare-13-01438-t003:** The relationships between emergency department admissions, ward admissions, intensive care unit admissions, survival, and BCKQ and mCCI scores were evaluated at the end of the follow-up period.

Scales	All Participants	Emergency Department Admissions			Ward Admissions			Intensive Care Unit Admissions			Survival Status		
	(n = 78) Median (Min–Max)	Admitted Median (Min–Max)	Not Admitted Median (Min–Max)	*p*	Admitted Median (Min–Max)	Not admitted Median (Min–Max)	*p*	Admitted Median (Min–Max)	Not Admitted Median (Min–Max)	*p*	Exitus Median (Min–Max)	Survived Median (Min–Max)	*p*
mCCI Total score	4 (1–8)	4 (1–8)	4 (2–7)	0.925	4 (1–7)	4 (1–8)	0.877	4 (3–6)	4 (1–8)	0.992	5 (2–8)	4 (1–7)	0.136
BCKQ Total score	23 (6–40)	25 (7–37)	20 (6–40)	0.005 *	26 (7–33)	23 (6–40)	0.150	23 (8–28)	23 (6–40)	0.744	20 (9–31)	23 (6–40)	0.504
1. Epidemiology	2 (0–4)	2 (0–4)	1 (0–4)	0.010 *	2 (1–4)	2 (0–4)	0.161	2 (0–4)	2 (0–4)	0.412	1.5 (0–2)	2 (0–4)	0.497
2. Etiology	3 (0–5)	3 (0–5)	2 (0–5)	0.033 *	3 (0–4)	3 (0–5)	0.729	3 (1–3)	3 (0–5)	0.855	2.5 (1–5)	3 (0–5)	0.596
3. Symptoms	2 (0–5)	2 (0–4)	2 (0–5)	0.318	2 (2–4)	2 (0–5)	0.042 *	2 (1–4)	2 (0–5)	0.303	2 (0–3)	2 (0–5)	0.607
4. Shortness of breath	2 (0–4)	2 (0–4)	1 (0–3)	0.042 *	2 (0–4)	2 (0–4)	0.144	3 (0–3)	2 (0–4)	0.406	1.5 (0–4)	2 (0–4)	0.708
5. Sputum	3 (0–5)	3 (0–5)	3 (0–5)	0.094	3 (1–5)	3 (0–5)	0.499	3 (0–4)	3 (0–5)	0.563	2.5 (1–4)	3 (0–5)	0.241
6. Infections	2 (0–4)	2 (0–4)	2 (0–3)	0.316	2.5 (0–5)	2 (0–5)	0.158	2 (0–2)	2 (0–5)	0.512	1 (1–3)	2 (0–5)	0.148
7. Exercise	2 (0–5)	2 (0–5)	2 (0–5)	0.053	2 (0–5)	2 (0–5)	0.243	3 (1–3)	2 (0–5)	0.422	2.5 (0–3)	2 (0–5)	0.878
8. Smoking	2 (0–4)	2 (0–3)	2 (0–4)	0.110	2 (0–3)	2 (0–4)	0.472	2 (2–2)	2 (0–4)	0.430	2 (1–3)	2 (0–4)	0.764
9. Vaccination	1 (0–4)	1 (0–4)	1 (0–4)	0.522	1 (0–3)	1 (0–4)	0.738	1 (0–3)	1 (0–4)	0.734	1 (0–3)	1 (0–4)	0.456
10. Inhalation BD	1 (0–4)	1 (0–4)	1 (0–3)	0.269	1 (0–3)	1 (0–4)	0.633	0 (0–3)	1 (0–4)	0.333	0 (0–2)	1 (0–4)	0.116
11. Antibiotics	2 (0–5)	2 (0–5)	2 (0–5)	0.758	2 (1–4)	2 (0–5)	0.953	1 (0–2)	2 (0–5)	0.019 *	1.5 (1–5)	2 (0–5)	0.541
12. Oral steroids	0 (0–4)	0 (0–3)	0 (0–4)	0.133	0 (0–3)	0 (0–4)	0.426	0 (0–0)	0 (0–4)	0.157	0 (0–3)	0 (0–4)	0.803
13. Inhaled steroids	0 (0–3)	0 (0–3)	0 (0–3)	0.048 *	0 (0–2)	0 (0–3)	0.717	0 (0–1)	0 (0–3)	0.887	0 (0–1)	0 (0–3)	0.576

BCKQ: Bristol COPD Knowledge Questionnaire, mCCI: modified Charlson Comorbidity Index, BD: Bronchodilators. * *p* < 0.05 statistically meaningful.

**Table 4 healthcare-13-01438-t004:** Risk factors affecting patients’ hospital readmissions in the first 6 months after discharge.

Characteristics (n = 78)	OR (95% CI)	*p*-Value
Age, years	1.00 (0.95–1.05)	0.975
COPD disease age, years	1.06 (1.00–1.13)	0.064
Educational status		
Primary school	Reference	-
Middle school	0.79 (0.21–2.95)	0.730
High school and above	2.04 (0.63–6.64)	0.238
Being a cared patient	1.90 (0.66–5.50)	0.237
Home oxygen device (LTOT)	1.79 (0.66–4.87)	0.253
Domiciliary NIV	2.60 (1.04–6.51)	0.041 *
BCKQ	1.10 (1.02–1.18)	0.010 *
mCCI	0.99 (0.72–1.38)	0.973
Comorbidity groups		
Cardiovascular diseases	1.36 (0.54–3.43)	0.522
Respiratory diseases	1.21 (0.48–3.04)	0.683
Metabolic diseases	0.82 (0.32–2.14)	0.691
Urorenal diseases	4.40 (1.42–13.67)	0.010 *

CI: Confidence interval; COPD: chronic obstructive lung disease; NIV: non-invasive ventilation; LTOT: long-term oxygen therapy. * *p* < 0.05 statistically meaningful.

## Data Availability

Our data cannot be shared openly, for protecting study participants’ privacy. If requested, the data will be shared for editors, reviewers, reviewers for statistical evaluation, for meta-analyses, and such other scientific reasons. You can access both the censored version of the study with anonymised personal data and the full dataset by contacting the corresponding author, Dr. Murat Yıldız. Contact details: email: drmuratyildiz85@gmail.com; phone number: +90 554 844 1305; affiliation: University of Health Sciences, Ankara Atatürk Sanatorium Training and Research Hospital, Department of Pulmonology.

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
