# Peer review of "The Influence of COPD Awareness on Hospital Admissions: A Paradoxical Relationship?†"

_healthcare, 2025, doi:10.3390/healthcare13121438_

Round 1
Reviewer 1 Report
Comments and Suggestions for Authors
Dear Authors (Editor),
First, I appreciated your planning and effort for this prospective study. I would like to express that it is a well-planned and well-written study. I hope that you will also carry out a study to measure the effect of disease education on hospital readmissions, which I think should be planned as a continuation of this study based on baseline knowledge.
1. I would like to question the lack of a cut-off value for the significance of the scores obtained with the BCKQ used in your study. Is there a cut-off value recommended both in the study developed by White et al.. and in the validation study conducted in your local language?
2. In your study including COPD D group patients, did you exclude readmissions for reasons other than COPD exacerbation?
Conclusion: Minor revision
I would like to re-evaluate after revision.
Author Response
Point by point response to reviewer 1
- I would like to question the lack of a cut-off value for the significance of the scores obtained with the BCKQ used in your study. Is there a cut-off value recommended both in the study developed by White et al. and in the validation study conducted in your local language?
Response: The original study and Turkish validation study have no cut-off value. In the ROC analysis conducted to evaluate the predictive performance of the total Bristol COPD Knowledge Questionnaire (BCKQ) score in relation to emergency department visits within six months after discharge, a cut-off value of 21 was identified. At this threshold, the sensitivity was calculated as 77.5%, specificity as 57.89%, positive predictive value (PPV) as 65.95%, and negative predictive value (NPV) as 70.97% (p = 0.0036, AUC = 0.683, 95% CI: 0.568–0.784).
Since the ROC analyses for the BCKQ total score did not yield statistically significant curves for the ward admissions or ICU admissions, these outcomes were not included in the final analysis. As shown in Table 3, the BCKQ total score was only significantly associated with emergency department visits.
No specific cut-off value for the BCKQ total score has been identified in the existing literature.
- In your study including COPD D group patients, did you exclude readmissions for reasons other than COPD exacerbation?
Response: Yes, for both the ward and ICU admissions, only hospitalisations due to COPD exacerbations were evaluated. Additionally, only emergency department visits related to COPD-related symptoms were considered and included in the study.
Reviewer 2 Report
Comments and Suggestions for Authors
- Sample Size: The study included a sample size of 78 cases. Although the authors mentioned that a post hoc analysis indicated the sample size was adequate, a larger sample size might enhance the reliability of the results, especially for a highly heterogeneous disease like COPD. It is recommended to further discuss the limitations of the sample size and its potential impact on the study outcomes in the discussion section.
- Exclusion Criteria: The exclusion criteria were relatively strict (e.g., excluding patients with other respiratory comorbidities), which may limit the generalizability of the findings. It is advisable to discuss how these exclusion criteria may affect the external validity of the study.
- Subgroup Analysis: Certain BCKQ subdomains (e.g., antibiotic knowledge) showed significant associations with ICU admission, but the clinical significance of these findings has not been fully explored. It is recommended to further investigate the relationship between these subdomains and specific clinical behaviors (e.g., antibiotic misuse or poor self-management).
- Knowledge-application gap:While the authors noted that increased knowledge did not directly improve clinical outcomes, the discussion on how to address this issue was insufficient. We recommend proposing specific intervention strategies (e.g., incorporating behavioral psychology approaches or enhancing practical skills training) to bridge this gap.
- Literature comparison:The comparison with other studies (e.g., Choi et al.) was relatively thorough, but further integration of inconsistent findings would be valuable—for instance, exploring whether discrepancies stem from cultural differences or healthcare system characteristics.
- Multivariable Analysis: Regarding the logistic regression, was the potential influence of confounding factors (e.g., disease severity) considered when selecting variables such as home NIV use and total BCKQ score? It is recommended to clarify whether multivariable adjustment or sensitivity analyses were conducted to address potential confounding.
- Interpretation of p-values: Some p-values are close to the significance threshold (e.g., the symptoms subdomain, p = 0.042), raising concerns about multiple comparisons. It is advisable to apply correction methods (e.g., Bonferroni correction) or clearly state that these findings should be interpreted as exploratory.
The English language in this paper is generally adequate, but requires correction of grammatical errors, optimization of word choice, and simplification of overly complex sentence structures. These improvements will significantly enhance readability and academic rigor.
Recommended Actions:
-
Systematically check subject-verb agreement and verb tenses;
-
Simplify lengthy sentences by avoiding interrupting clauses;
-
Standardize terminology and eliminate redundant expressions.
Below are the sections requiring corrections
Main Issues
(1) Grammatical Errors
Subject-Verb Agreement
Original Text:
"Data on emergency department visits, hospitalizations, and intensive care unit (ICU) admissions were collected for a six-month follow-up period."
Issue: Although "data" is typically treated as plural in academic writing, it can also be considered singular in American English (especially when referring to a dataset). It is recommended to consistently use the plural form ("data were") or adjust according to the journal's style.
Inconsistent Tense
Original Text:
"The Bristol COPD Knowledge Questionnaire (BCKQ) was administered prior to discharge, and patients are followed for six months."
Issue: The first clause uses the past tense ("was"), while the second clause uses the present tense ("are"), creating tense inconsistency.
Suggested Revision:
"The BCKQ was administered prior to discharge, and patients were followed for six months."
(2) Inappropriate Word Choice
Inaccurate Terminology
Original Text:
"Patients with neurological diseases such as Alzheimer’s, dementia, or Parkinson’s were excluded."
Issue: "Dementia" is a broad term that encompasses Alzheimer’s disease, so listing them together creates logical overlap.
Suggested Revision:
"Patients with neurological diseases (e.g., Alzheimer’s, Parkinson’s) or other forms of dementia were excluded."
Redundant Expression
Original Text:
"The study was performed in accordance with the Helsinki Declaration and was approved by the local ethics committee."
Issue: Excessive use of passive voice makes the sentence wordy.
Suggested Revision:
"The study adhered to the Helsinki Declaration and received approval from the local ethics committee."
(3) Sentence Structure Issues
Confusing Long Sentences
Original Text:
"In our study, the increased frequency of emergency department visits, despite higher knowledge levels, indicates that patients may struggle to effectively apply their knowledge."
Issue: The parenthetical phrase ("despite higher knowledge levels") interrupts the main clause, reducing readability.
Suggested Revision:
"Despite higher knowledge levels, our study found increased emergency visits, suggesting patients struggle to apply knowledge effectively."
2. Other Language Issues
Missing Articles
Original Text:
"Patients were monitored via telephone follow-up, face-to-face interviews, or through hospital’s data recording system."
Issue: The definite article ("the") is missing before "hospital’s."
Suggested Revision:
"Patients were monitored via telephone follow-up, face-to-face interviews, or through the hospital’s data recording system."
Preposition Misuse
Original Text:
"Patients residing outside the central districts of Ankara were excluded."
Issue: Although grammatically correct, "outside" could be misinterpreted as "non-central districts" (including suburbs). For greater precision, consider:
Suggested Revision:
"Patients residing beyond the central districts of Ankara were excluded."
3. Suggestions for Academic Writing Style
Avoid Colloquial Expressions
Original Text:
"This finding is one of the most important results of this study."
Suggested Revision:
"This finding represents a key contribution of the study."
Passive vs. Active Voice
Original Text (Passive):
"The questionnaire was administered by researchers."
Suggested Revision (Active, More Concise):
"Researchers administered the questionnaire."
4. Example Revision (Excerpt from Abstract)
Original Text:
"This study aims to investigate whether COPD knowledge levels influence hospital readmissions and healthcare utilization in patients hospitalized for COPD exacerbations."
Revised:
"This study examined the association between COPD knowledge levels and healthcare utilization (including hospital readmissions) in patients hospitalized for acute exacerbations."
Revisions made because:
- "Aims to investigate" → "Examined" (study completed, using past tense)
- Added parentheses to clarify what "healthcare utilization" includes
- Changed "COPD exacerbations" to more precise "acute exacerbations"
Author Response
- Question 1. Sample Size: The study included a sample size of 78 cases. Although the authors mentioned that a post hoc analysis indicated the sample size was adequate, a larger sample size might enhance the reliability of the results, especially for a highly heterogeneous disease like COPD. It is recommended to further discuss the limitations of the sample size and its potential impact on the study outcomes in the discussion section.
Response: Sample Size Considerations:
One of the acknowledged limitations of this study is the relatively small sample size (n = 78). While this number may seem modest, a post hoc power analysis confirmed that the sample was statistically adequate to detect meaningful differences in the studied outcomes. Nevertheless, given the clinical and behavioral heterogeneity of COPD patients, a larger cohort could have enhanced the robustness and external validity of the findings. Future studies with broader inclusion may provide more generalizable insights and enable subgroup analyses that were not feasible within the current sample size.
- Question 2. Exclusion Criteria: The exclusion criteria were relatively strict (e.g., excluding patients with other respiratory comorbidities), which may limit the generalizability of the findings. It is advisable to discuss how these exclusion criteria may affect the external validity of the study.
Response: External Validity and Exclusion Criteria
The study employed relatively strict exclusion criteria, such as the omission of patients with significant comorbid respiratory or neurological conditions, in order to minimize confounding and ensure a more homogeneous sample focused specifically on COPD-related hospitalizations. While this approach strengthens internal validity and enhances the clarity of observed associations, it may limit the external generalizability of the findings to the broader COPD population, where multimorbidity is common. Nonetheless, this methodological choice was necessary to isolate the effects of disease-specific knowledge and avoid misinterpretation due to overlapping pathologies. Future studies may benefit from a more inclusive design that reflects real-world clinical diversity while still controlling for key confounders through statistical adjustments.
- Question 3. Subgroup Analysis: Certain BCKQ subdomains (e.g., antibiotic knowledge) showed significant associations with ICU admission, but the clinical significance of these findings has not been fully explored. It is recommended to further investigate the relationship between these subdomains and specific clinical behaviors (e.g., antibiotic misuse or poor self-management).
Response: Antibiotic Knowledge and ICU Admission
In our study, patients with higher scores in the antibiotic subdomain of the BCKQ were less likely to be admitted to the intensive care unit (ICU) within six months following discharge. This finding suggests that improved knowledge regarding antibiotic use may play a protective role by enabling patients to recognize infection-related symptoms earlier and seek timely outpatient care, thereby preventing progression to severe exacerbations. This aligns with the 2021 Global Initiative for Chronic Obstructive Lung Disease (GOLD) guidelines, which recommend prescribing antibiotics for patients exhibiting increased sputum purulence alongside dyspnea or changes in sputum volume, emphasizing the importance of early recognition and treatment of bacterial infections to prevent severe outcomes . Furthermore, a retrospective study by Vanoverschelde et al. (2023) found that in-hospital antibiotic use among patients with severe acute exacerbations of COPD was associated with longer hospital stays, highlighting the significance of appropriate antibiotic management in influencing patient outcomes . These findings support the hypothesis that focused knowledge, particularly on infection management, can have a tangible impact on disease trajectory and healthcare utilization in COPD.
(Vanoverschelde A, Van Hoey C, Buyle F, Den Blauwen N, Depuydt P, Van Braeckel E, Lahousse L. In-hospital antibiotic use for severe chronic obstructive pulmonary disease exacerbations: a retrospective observational study. BMC Pulm Med. 2023 Apr 25;23(1):138. doi: 10.1186/s12890-023-02426-3. PMID: 37098509; PMCID: PMC10127022.)
- Question 4. Knowledge-application gap:While the authors noted that increased knowledge did not directly improve clinical outcomes, the discussion on how to address this issue was insufficient. We recommend proposing specific intervention strategies (e.g., incorporating behavioral psychology approaches or enhancing practical skills training) to bridge this gap.
Response: Bridging the Knowledge-Behavior Gap
Although increased disease knowledge is generally considered beneficial, our findings support the growing consensus that knowledge alone may be insufficient to produce meaningful behavior change in chronic disease management. This discrepancy highlights the so-called “knowledge-behavior gap” in COPD care. Patients may be aware of appropriate actions but still fail to implement them due to psychological, cognitive, or social barriers. As such, future interventions should not only focus on education but also incorporate behavioral change techniques, such as motivational interviewing, goal-setting, problem-solving training, and practical self-management skill workshops. Additionally, emotion regulation strategies, such as stress management or coping skills training, may be especially useful for patients who experience anxiety in response to symptom exacerbation. Embedding such elements into structured pulmonary rehabilitation or nurse-led follow-up programs could enhance patients’ ability to translate knowledge into action, ultimately improving long-term outcomes.
- Question 5. Literature comparison:The comparison with other studies (e.g., Choi et al.) was relatively thorough, but further integration of inconsistent findings would be valuable—for instance, exploring whether discrepancies stem from cultural differences or healthcare system characteristics.
Response: While Choi et al. reported a positive correlation between disease-specific knowledge and improved clinical outcomes in COPD patients, our findings revealed a seemingly paradoxical trend, wherein higher knowledge scores were associated with increased healthcare utilization. This discrepancy may be attributed to contextual and systemic differences. For instance, Choi's study was conducted in a structured healthcare environment with well-integrated COPD action plans and follow-up systems, which likely facilitated the translation of knowledge into appropriate behavior. In contrast, in settings where access to continuity of care, self-management training, or primary care support is more fragmented—as is often the case in resource-limited health systems—patients may resort to emergency services more frequently despite having sufficient knowledge. Additionally, cultural differences in health-seeking behavior, trust in outpatient services, and perceived urgency of symptoms may influence how knowledge is applied. These contextual factors should be considered when interpreting cross-cultural comparisons in COPD management research.
- Question 6. Multivariable Analysis: Regarding the logistic regression, was the potential influence of confounding factors (e.g., disease severity) considered when selecting variables such as home NIV use and total BCKQ score? It is recommended to clarify whether multivariable adjustment or sensitivity analyses were conducted to address potential confounding.
Response: Variable Selection and Confounding Control in Logistic Regression
In our regression analysis, we utilized the enter method to include variables based on both clinical relevance and theoretical considerations. Since all participants were classified as GOLD Group D, the severity of COPD was consistent across the cohort, thereby minimizing variability in disease stage as a potential confounder. Variables included in the model—such as age, comorbid conditions (as measured by the mCCI), use of domiciliary non-invasive ventilation (NIV), and long-term oxygen therapy—were selected due to their established association with hospital readmission risk. Additionally, knowledge-related parameters, including the total BCKQ score, were incorporated to address the primary objective of the study. Although no formal multivariable screening process (e.g., stepwise selection) was used, care was taken to construct a model that accounted for key clinical and behavioral contributors without overfitting.
- Question 7. Interpretation of p-values: Some p-values are close to the significance threshold (e.g., the symptoms subdomain, p = 0.042), raising concerns about multiple comparisons. It is advisable to apply correction methods (e.g., Bonferroni correction) or clearly state that these findings should be interpreted as exploratory.
Response: Thank you for this insightful comment. We agree that multiple comparisons across the BCKQ subdomains and clinical outcomes (including emergency department visits, general ward admissions, and ICU admissions) raise the possibility of inflated type I error. While several subdomains showed p-values below the conventional threshold (p < 0.05), none of them remained statistically significant after applying the Bonferroni correction (α ≈ 0.0038 for 13 comparisons per outcome category).
However, as our study aimed to explore potential patterns and generate hypotheses regarding disease-specific knowledge and healthcare utilization, we consider these findings exploratory in nature. We have explicitly acknowledged this limitation and added the following sentence to the Discussion section of the manuscript:
"Given the exploratory nature of this study, the associations observed between BCKQ subdomains and clinical outcomes (including emergency visits, general ward admissions, and ICU admissions) should be interpreted with caution. Although several subdomains reached nominal significance (p < 0.05), none remained statistically significant after applying a conservative Bonferroni correction. Therefore, these findings should be considered hypothesis-generating and warrant further investigation in larger, confirmatory studies."
We hope this clarification addresses your concern appropriately.
- Question 8. Comments on the Quality of English Language
The English language in this paper is generally adequate, but requires correction of grammatical errors, optimization of word choice, and simplification of overly complex sentence structures. These improvements will significantly enhance readability and academic rigor.
Response: Thank you for your valuable feedback regarding the language and writing quality of the manuscript. In response to your suggestions, we have thoroughly reviewed the entire text to improve grammar, simplify overly long and complex sentences, correct issues with subject-verb agreement and tense consistency, and ensure appropriate use of academic vocabulary. We also removed redundant expressions and restructured selected paragraphs to improve clarity and flow. These revisions were carefully implemented to enhance both the readability and the scholarly tone of the manuscript. We sincerely appreciate your guidance in strengthening the linguistic and stylistic quality of our work.
Reviewer 3 Report
Comments and Suggestions for Authors
The article tacles an important topic on patients' knowledge about COP and the influence of them on hospital admissions. The major COPD comorbidities were exluded suggesting, that the authors seaked more pure COPD, however some comorbidities were included.
The compliance to the treatment was not assessed, in my opinion the correlation with the knowledge on COPD and compliance could be an advantage. No information about the access to selvmanagement tools or availability to acut primary care access was provided.
I suggest to include reassessment of patients according to the recent GOLD classification and provide insights for the future research in conclusions.
Author Response
Reviewer Comment 1:
"The major COPD comorbidities were excluded suggesting that the authors sought a more pure COPD, however, some comorbidities were included."
Author Response:
Thank you for this insightful observation. We agree that the exclusion of certain comorbidities while allowing others warrants clarification. Asthma was specifically excluded in our study due to its potential to confound both the clinical course and patient knowledge patterns in COPD, especially in patients with asthma-COPD overlap (ACO). Including such cases could have biased the results by introducing distinct disease perceptions, treatment responses, and educational needs. Other stable comorbidities (e.g., hypertension, diabetes) were not excluded, as they reflect the real-world burden commonly observed in COPD Group D populations and were accounted for via the Charlson Comorbidity Index (CCI).
Reviewer Comment 2:
"The compliance to the treatment was not assessed; in my opinion, the correlation with the knowledge on COPD and compliance could be an advantage."
Author Response:
We appreciate this valuable suggestion. Indeed, exploring the relationship between disease-specific knowledge and treatment compliance would have enriched our findings. Unfortunately, due to the scope and design limitations of this study, treatment adherence could not be objectively evaluated. Nevertheless, we recognize the relevance of this issue and recommend it as a focal point for future research. And we added to the study this paragraph :
“Future studies should aim to assess not only disease-specific knowledge but also patient adherence to COPD therapies, as knowledge alone may not ensure effective disease self-management. Integrating validated adherence measures could help uncover whether higher knowledge levels translate into better treatment compliance and reduced healthcare utilization. Such multidimensional approaches may offer deeper insights into the behavioral pathways influencing clinical outcomes in COPD.”
Reviewer Comment 3:
"No information about the access to self-management tools or availability to acute primary care access was provided."
Author Response:
Thank you for this important point. We have addressed this issue in the revised version of the manuscript with the following paragraph added to the Discussion section:
“In Turkey, the absence of a strict referral chain within the healthcare system allows patients to directly access secondary or tertiary care, particularly specialists such as pulmonologists. Many patients prefer to bypass primary care, believing that a specialist is more likely to understand their condition accurately. Additionally, diagnostic and treatment capacities at the primary care level remain limited in many regions, leading to a reliance on hospital-based care. Regarding self-management, patients were prescribed home-based treatments such as nebulizers, long-term oxygen therapy (LTOT), or non-invasive ventilation (NIV) when clinically indicated. Moreover, they were encouraged to acquire pulse oximeters for home monitoring. Prior to discharge, all patients received brief face-to-face training on managing acute exacerbations at home, which included recognizing warning signs and adjusting treatments accordingly.”
Reviewer Comment 4:
"I suggest to include reassessment of patients according to the recent GOLD classification and provide insights for the future research in conclusions."
Author Response:
Thank you for this constructive recommendation. As the study population was composed entirely of GOLD Group D patients according to the classification available at the time of data collection, no reclassification was performed under the newly updated GOLD 2023 system. However, we acknowledge the shift in GOLD's emphasis toward symptom burden and exacerbation risk rather than ABCD grouping. This paradigm shift further reinforces the need for integrated approaches targeting both clinical and behavioral dimensions of COPD management. Future studies might benefit from stratifying patients using newer multidimensional indices that align with the updated GOLD criteria to better capture phenotypic variability and tailor interventions accordingly.
Reviewer 4 Report
Comments and Suggestions for Authors
Overall, an interesting draft highlighting the potential association between knowledge levels of patients with COPD with exacerbation frequency, hospital readmissions, or hospitalization rates. This study asked a reasonable question and was conducted in a feasible manner. Many of the references (8/19) are from before 2018, indicating a lack of updated information. Therefore, the authors should consider adding more current references. Nevertheless, the manuscript’s research question is interesting, however, it needs revision in terms of writing style and data presentation. Specifically, I invite the authors to clarify/revise the following issues.
- I think the introduction requires restructuring to improve the flow, and provide more details regarding the subject, for example, from the more general what is COPD, what does hospitalizations means for patients with COPD, what factors contribute to hospitalizations, to the lack in the literature, and consequently the need for the study.
- Regarding the introduction, I would like a greater explanation regarding why this study was necessary and how this study differentiates from other similar ones.
- The methods sections could benefit by clearing a few things. First, how long did it take to complete the questionnaires. Second, more clearly describe the post-hoc power analysis or maybe a sample size calculation? Third, how did you interview the patients, how were they approached, please more clearly describe the process of data collection? Fourth, how was anonymity/confidentiality maintained?
- The results section along with the tables look fine.
- The discussion section could be expanded by dedicating a paragraph for each major finding based on your aims (COPD and factors related to hospitalization) by comparing it with existing literature, and by giving a potential explanation for each one. Towards that end I strongly suggest to the authors to add a higher number of relevant references to further cement their main findings to the literature.
- The limitations could be expanded, for example other variables could factor in the hospitalization rates for example adherence to COPD medication.
- Finally, the conclusion section could be expanded and focus on the main findings, implications, and recommendations of this study.
Author Response
- Reviewer comment 1: I think the introduction requires restructuring to improve the flow, and provide more details regarding the subject, for example, from the more general what is COPD, what does hospitalizations means for patients with COPD, what factors contribute to hospitalizations, to the lack in the literature, and consequently the need for the study.
Response: Thank you very much for your constructive feedback regarding the Introduction. In response, we have revised and expanded the introductory section to enhance the logical flow from a general overview of COPD and its burden to the specific gap in the literature that justifies our research. We retained our original structure but integrated additional contextual paragraphs to improve coherence and depth, particularly focusing on the role of disease knowledge and its potential impact on outcomes such as readmissions. Furthermore, we have added four up-to-date references (2020–2024) to support and strengthen the revised narrative.
These revisions aim to clarify the rationale for our study and emphasize its originality in assessing baseline knowledge levels prior to any educational intervention.
INTRODUCTION:
Chronic obstructive pulmonary disease (COPD) is among the leading causes of chronic morbidity and mortality worldwide. The disease is characterized not only by persistent airflow limitation but also by a spectrum of systemic effects and exacerbation patterns that require frequent interactions with the healthcare system. As COPD progresses, patients may experience an increasing number of acute episodes that disrupt their daily lives and often necessitate urgent medical attention.
Chronic obstructive pulmonary disease (COPD) is a progressive respiratory condition characterized by frequent exacerbations, emergency department visits, and hospitalizations, all of which contribute to a substantial global health burden. These acute exacerbations not only diminish patients' quality of life but also significantly increase the risk of all-cause mortality, particularly in individuals with moderate to severe COPD. Evidence suggests that frequent exacerbations accelerate lung function decline, intensify symptom severity, and lead to greater reliance on healthcare resources (1, 2). High readmission rates following hospitalizations further exacerbate this burden, creating an unsustainable strain on healthcare systems. For patients, recurrent hospitalizations are both physically taxing and emotionally distressing, often reflecting suboptimal disease management. As such, identifying the factors that contribute to exacerbations is crucial for improving patient outcomes and reducing healthcare costs (3).
While pharmacologic treatments and guideline-directed care are central to managing COPD, recent studies underscore the critical role of patient knowledge and self-management capacity in determining outcomes. Patients who better understand their disease are more likely to adhere to maintenance therapy, recognize early warning signs of exacerbation, and seek timely care. However, existing literature also indicates substantial variability in COPD patients’ baseline knowledge, and how this translates into clinical outcomes remains poorly understood.( Bourbeau J, Farias R, Li PZ, Gauthier G, Battisti L, Chabot V, Beauchesne MF, Villeneuve D, Côté P, Boulet LP. The Quebec Respiratory Health Education Network: Integrating a model of self-management education in COPD primary care. Chron Respir Dis. 2018 May;15(2):103-113. doi: 10.1177/1479972317723237. Epub 2017 Jul 27. PMID: 28750556; PMCID: PMC5958467. Bischoff EWMA, Ariens N, Boer L, Vercoulen J, Akkermans RP, van den Bemt L, Schermer TR. Effects of Adherence to an mHealth Tool for Self-Management of COPD Exacerbations. Int J Chron Obstruct Pulmon Dis. 2023 Nov 1;18:2381-2389. doi: 10.2147/COPD.S431199. PMID: 37933244; PMCID: PMC10625742. Fischer C, Siakavara M, Alter P, Vogelmeier CF, Speicher T, Pott H, Watz H, Bals R, Trudzinski F, Herth F, Ficker JH, Wagner M, Lange C, Stoycheva K, Randerath W, Behr J, Fähndrich S, Welte T, Pink I, Kahnert K, Seeger W, Kuhnert S, Gessler T, Adaskina N, Jörres RA. Association of Patients' Knowledge on the Disease and Its Management with Indicators of Disease Severity and Individual Characteristics in Patients with Chronic Obstructive Pulmonary Disease (COPD): Results from COSYCONET 2. Patient Prefer Adherence. 2024 Dec 2;18:2383-2393. doi: 10.2147/PPA.S488165. PMID: 39650575; PMCID: PMC11624521.)
Evaluating COPD patients' current level of knowledge about their disease is a vital step toward developing effective educational programs. This study aims to determine whether the baseline knowledge levels of patients—specifically those who have not received prior intervention or education regarding COPD—are associated with exacerbation frequency, hospital readmissions, or hospitalization rates. Previous research highlights variability in patients’ understanding of their condition, which may influence health outcomes (4, 5, 6, 7, 8). By exploring the relationship between knowledge levels and exacerbation frequency, this study seeks to identify critical knowledge gaps that may contribute to suboptimal disease management. Gaining insights into these baseline knowledge deficits will be instrumental in designing tailored educational interventions aimed at reducing exacerbations and improving the overall management of COPD.
Unlike many previous studies that focused on patients already enrolled in structured education programs or pulmonary rehabilitation, our study uniquely evaluates patients at a point when no formal education has yet been delivered. This design offers a clearer lens to investigate how unassisted knowledge alone may relate to healthcare utilization in real-world settings. Our findings may inform the development of more targeted, personalized interventions that align with actual patient needs and learning baselines. ( Mongiardo MA, Robinson SA, Finer EB, Cruz Rivera PN, Goldstein RL, Moy ML. The Effect of a web-based physical activity intervention on COPD knowledge: A secondary cohort study. Respir Med. 2021 Dec;190:106677. doi: 10.1016/j.rmed.2021.106677. Epub 2021 Nov 8. PMID: 34775350; PMCID: PMC8710703. )
- Reviewer comment 2: Regarding the introduction, I would like a greater explanation regarding why this study was necessary and how this study differentiates from other similar ones.
Response : Thank you again for your thoughtful comment. In response, we have revised the final part of the Introduction to clarify the necessity and originality of our study. Specifically, we emphasize that our study uniquely focuses on patients who have not yet received formal COPD education. This design allows us to explore the direct relationship between unassisted disease knowledge and healthcare utilization outcomes. We believe this approach provides a novel perspective compared to prior studies that primarily included pre-educated or rehabilitated patients. We hope this addition effectively addresses your concern.
- Reviewer comment 3: The methods sections could benefit by clearing a few things. First, how long did it take to complete the questionnaires. Second, more clearly describe the post-hoc power analysis or maybe a sample size calculation? Third, how did you interview the patients, how were they approached, please more clearly describe the process of data collection? Fourth, how was anonymity/confidentiality maintained?
Response:
Survey Completion Time
The administration of the Bristol COPD Knowledge Questionnaire (BCKQ) was conducted in a face-to-face format by trained healthcare professionals prior to patient discharge. On average, patients completed the questionnaire in approximately 15 to 20 minutes, depending on their cognitive capacity and level of engagement. All sessions were conducted in a quiet setting to minimize distractions and ensure accurate responses.
Post-hoc Power Analizi / Örneklem Büyüklüğü
A post-hoc power analysis was performed using the software G*Power version 3.1.9.7 to evaluate the statistical sensitivity of the logistic regression model employed in this study. Based on an observed effect size (odds ratio), a significance level (α) of 0.05, and a total sample size of 78, the calculated statistical power (1–β) exceeded 0.80, indicating an acceptable probability of detecting true effects. While no a priori sample size estimation was conducted due to the observational design, the post-hoc analysis supports the adequacy of the sample size for the primary outcome variables examined.
Data Collection and Patient Approach Process
Patients hospitalized for COPD exacerbation were identified through hospital admission logs and approached during their inpatient stay. After obtaining informed consent, baseline demographic data and clinical characteristics were collected through structured interviews and medical record review. The BCKQ was administered in person by the research team prior to discharge. Follow-up data on emergency visits, readmissions, and ICU stays were obtained from the hospital information system and supplemented by scheduled telephone interviews conducted at three and six months post-discharge.
Anonymity and Privacy
Ethical approval for this study was obtained from the institutional review board of Keçiören Training and Research Hospital Clinical Research Ethics Committee (approval date: April 13, 2021; decision number: 2012-KEAK 15/2245), and all participants provided written informed consent. Patient anonymity and confidentiality were strictly maintained throughout the study. Data were coded and stored in a secure, password-protected electronic database accessible only to authorized study personnel. Identifiable information was removed prior to data analysis to ensure compliance with data protection regulations and ethical standards.
- Reviewer comment 4: The results section along with the tables look fine.
Response: thank you for your valueble comment
- Reviewer comment 5: The discussion section could be expanded by dedicating a paragraph for each major finding based on your aims (COPD and factors related to hospitalization) by comparing it with existing literature, and by giving a potential explanation for each one. Towards that end I strongly suggest to the authors to add a higher number of relevant references to further cement their main findings to the literature.
Response: Thank you for your constructive suggestion. In response, we have revised the discussion section to structure it more explicitly around the main findings of our study. Each major outcome—such as the relationship between baseline knowledge and emergency admissions, ICU admissions, and hospital readmissions—has now been addressed in separate, clearly delineated paragraphs. In each case, we included comparisons with relevant findings from the literature and discussed plausible interpretations. Additionally, we enriched the discussion with more up-to-date and diverse references to support and contextualize our results more comprehensively. We hope that these revisions enhance the clarity and academic rigor of the discussion section.
- Reviewer comment 6: The limitations could be expanded, for example other variables could factor in the hospitalization rates for example adherence to COPD medication.
Response: Thank you for your insightful remark. We agree that treatment adherence is a critical factor influencing hospitalization risk. While our study did not directly assess medication adherence, this limitation has now been acknowledged in the manuscript. Furthermore, we have added a sentence in the discussion highlighting that future studies should incorporate validated measures of adherence to better understand the interaction between knowledge, behavior, and clinical outcomes.
- Reviewer comment 7: Finally, the conclusion section could be expanded and focus on the main findings, implications, and recommendations of this study.
Response: We appreciate your valuable suggestion. In response, we have revised the conclusion section to better summarize the key findings of our study and to articulate their implications for both clinical practice and future research. The updated conclusion now emphasizes the potential utility of baseline knowledge assessment in guiding tailored educational strategies and highlights the need for multidimensional interventions that integrate behavioral and structural factors.

Reviewer 5 Report
Comments and Suggestions for Authors The study is methodologically robust in several ways. Utilizing the validated BCKQ tool (including a Turkish version) enhances the measurement of patient knowledge. The exclusion criteria are clearly defined, and the six-month follow-up period, while somewhat brief, is justifiable given the frequency of exacerbations in Group D COPD patients. However, some methodological aspects require critical reflection. The relatively small sample size (n=78), while statistically powered post hoc, may limit the external validity of the findings. Additionally, although the study thoroughly assesses knowledge, it overlooks variables such as anxiety, depression, self-efficacy, and health literacy—all known mediators in the relationship between knowledge and behavior. This omission restricts the ability to investigate more complex pathways behind the observed associations.The statistical analyses are generally appropriate, and the use of logistic regression to identify predictors of hospital readmissions adds depth to the findings. However, it is important to note that the conclusions drawn from these analyses rely heavily on correlations that may not indicate causation. For example, higher knowledge may not directly lead to increased hospital admissions. Instead, it may serve as a proxy for other unmeasured factors, such as patient vigilance, access to care, or heightened health anxiety. While the discussion acknowledges some of these possibilities, it could benefit from a more detailed exploration of the potential psychological burden of knowledge and the role of behavioral activation or maladaptive coping strategies.
One of the manuscript’s notable contributions is its emphasis on the limitations of traditional education-based interventions. By highlighting the disjunction between knowledge acquisition and behavior change, the study aligns with a growing body of literature that emphasizes the need for integrative approaches combining information with skill-building, emotional regulation, and personalized support. This recognition has practical implications for healthcare systems aiming to reduce readmissions and optimize resource allocation.
Tables are generally effective, although Table 3 is particularly data-dense and may benefit from visual reorganization or supplementary figures. The presentation of BCKQ subdomains in relation to various outcome measures is thorough. However, some statistically significant findings (e.g., the link between antibiotic knowledge and ICU admission) remain difficult to interpret causally. The discussion attempts to contextualize these findings but sometimes veers into speculative territory without adequate theoretical grounding.
The manuscript is well-referenced, utilizing a variety of international studies to place its findings within a broader context. The cited works encompass different populations and healthcare settings, which helps underscore the variability in how knowledge impacts outcomes across cultures and systems.
In conclusion, the manuscript presents a valuable and thought-provoking perspective on the complexities of patient education in COPD. While the results may seem counterintuitive, they are a compelling reminder that information alone is not a cure-all. The findings urge clinicians and policymakers to adopt a more holistic view of what defines effective patient engagement and to acknowledge the multifactorial nature of healthcare utilization. With further exploration of the psychological and behavioral aspects of patient response, along with a clearer articulation of the study’s limitations, the paper would significantly contribute to the ongoing conversation regarding chronic disease self-management.
Author Response
- Reviewer comment 1: Sample Size and Methodology:
We acknowledge that the sample size was modest; however, as noted by the reviewer, the study was carefully designed, and a post-hoc power analysis confirmed adequate statistical power. This has been explicitly stated in the revised Methods section.
- Reviewer comment 2: Psychosocial Mediators (e.g., Anxiety, Self-efficacy, Health Literacy):
We agree that variables such as anxiety, depression, self-efficacy, and health literacy may mediate the relationship between knowledge and behavior. While these were not assessed in the present study, we have acknowledged this limitation in the revised Discussion and emphasized the need for future studies to adopt a multidimensional approach incorporating these factors.
- Reviewer comment 3: Psychological Burden of Knowledge:
We found the reviewer’s interpretation particularly insightful. In response, we have expanded the discussion to explore how heightened knowledge might be associated with increased health anxiety or behavioral hypervigilance, potentially leading to greater healthcare utilization. Relevant literature was cited to support this interpretation.
- Reviewer comment 4: BCKQ Subdomain Interpretation:
We appreciate the comment regarding the complexity of interpreting individual BCKQ subdomain results. We have revised the language in the Results and Discussion sections to present these findings more cautiously and framed them as exploratory, in line with the study’s scope.
- Reviewer comment 5: Table 3 Presentation:
Thank you for this thoughtful suggestion. While we agree that Table 3 presents a high volume of information, we believe that maintaining all key findings in a single, consolidated table enables a clearer overview and facilitates interpretation of the associations across multiple BCKQ subdomains and outcomes. Breaking the table into separate components may compromise the ability to compare results side by side, which we consider essential for readers to fully grasp the pattern of relationships. For this reason, we have retained the original structure but have ensured that the formatting remains as clean and reader-friendly as possible.
- Reviewer comment 6: Literature and Cultural Framing:
We are grateful for the acknowledgment of our efforts to situate the findings within a diverse international literature. We have further strengthened this by clarifying how healthcare system structure and patient behavior may differ across cultural contexts.
Round 2
Reviewer 4 Report
Comments and Suggestions for Authors
I would like to comment the authors for adequately, improving their manuscript, they did a job well done in addressing my concerns.